# Comparative venomics suggests an evolutionary adaption of spider venom from predation to defense
Tim Lüddecke [1,2] ✉, Sabine Hurka [1,2,3], Josephine Dresler[1,2], Thomas Lübcke[4], Volker von Wirth[5], Günter Lochnit [6], Thomas Timm[6], Volker Herzig [7,8] & Andreas Vilcinskas[1,2,9]

Most spiders deploy paralytic venom for prey capture, but adults of the Nurse's thorn finger (*Cheiracanthium punctorium*) instead produce a predominantly defensive venom to safeguard their offspring. Here, we characterize the molecular repertoire of *C. punctorium* venom to shed light on its evolutionary history. Unlike venom in other spiders, *C. punctorium* venom mostly comprises neurotoxic double-domain neurotoxin 19 family (CSTX) peptides and enzymes, such as phospholipase $A_2$ (PLA$_2$). Comparative venomics in four spiders representing two infraorders shows that CSTXs arise following the mygalomorph–araneomorph split ~300 mya by means of ancestral gene duplication and functional specialization. A gene fusion event then appeared to have merged CSTXs from two distinct clades to form the double-domain toxin. PLA$_2$ proteins are convergently recruited to *C. punctorium* to fulfil a defensive function and are strikingly similar to proalgesic PLA$_2$ proteins in bee venom. These complex, multimodal molecular innovations in venom systems highlight nature's tendency to use the same molecular solutions for similar ecological challenges across diverse animal lineages.

The independent acquisition of similar traits in unrelated organisms is known as convergent evolution[1]. Animal venoms offer a compelling example, given their biological importance, direct genotype–phenotype connectivity, medical relevance, and cultural influence[2–5]. They serve three primary functions: hunting, intraspecific competition, and anti-predator defense[6]. The latter is medically relevant because defensive envenomation claims thousands of human lives per year, mainly due to snakebite[7]. A hallmark symptom of defensive venoms is immediate, often excruciating, and long-lasting pain, which causes predators to abort their attack and thus safeguards the defensive organism[8–10]. The effects of envenomation also trigger learning, leading to avoidance behavior in predators, and conferring a selective advantage to the defender[8]. Defensive venoms are thus effective weapons that underpin the evolutionary success of many animals and drive the predator–prey arms race[8].

If measured in biodiversity, evolutionary age, and diversity of habitats conquered, spiders are the most successful order of venomous animals[11].

Since their emergence during the Ordovician period, spiders have spread to all continents except Antarctica, and comprise more than 53,000 extant species distributed across three infraorders (Mesothelae, Mygalomorphae, and Araneomorphae)[11,12]. All spider families except one carry functional venom systems that are used to overpower insect prey[11]. Their venoms may contain thousands of neurotoxic peptides, enzymes, and small organic compounds, with neurotoxins as the major components[13,14]. These are small peptides characterized by disulfide-bonds often forming a pseudoknot motif known as the inhibitor cysteine knot (ICK). An array of different ICK toxin families are known, with famous examples being neurotoxin 10 (huwentoxins, Hwtx), neurotoxin 27 (jingzhaotoxins, Jztx), and neurotoxin 19 (*Cupiennius salei* toxins, CSTX) that are distinguished by their arrangement and number of disulfide bonds forming the name-giving motif. In some spiders, they appear as double-domain toxins with bivalent activity[15,16]. Spider venom components, especially ICK toxins, have strong translational potential in the field of medicine, but little is known about their evolution and ecology[17].

[1]Branch for Bioresources, Fraunhofer Institute for Molecular Biology and Applied Ecology, Gießen, Germany. [2]LOEWE-Centre for Translational Biodiversity Genomics, Frankfurt am Main, Germany. [3]BMBF Junior Research Group in Bioeconomy (BioKreativ) "SymBioÖkonomie", Giessen, Germany. [4]Senckenberg Museum für Naturkunde, Görlitz, Germany. [5]Theraphosid Research Team, Eitting, Germany. [6]Institute for Biochemistry, Justus Liebig University of Gießen, Gießen, Germany. [7]Centre for Bioinnovation, University of the Sunshine Coast, Sippy Downs, QLD, Australia. [8]School of Science, Technology and Engineering, University of the Sunshine Coast, Sippy Downs, QLD, Australia. [9]Institute for Insect Biotechnology, Justus Liebig University of Gießen, Gießen, Germany. ✉e-mail: tim.lueddecke@ime.fraunhofer.de

In contrast to other venomous animals, almost all spiders use their venom predominantly for hunting, so defensive bites do usually not evoke the defense-related symptoms of envenomation[8,11]. Albeit pain is often reported after spider bites, these symptoms are usually mild and short-lived[18]. In addition, a variety of defensive strategies evolved within the spider kingdom to reduce or avoid venom utilization in defensive context[11,19,20]. Examples of algogenic spider toxins include vanillotoxins or the peptide Hm1a from theraphosids[16,21,22]. The most well-studied example of defensive toxins is produced by male atracids, which have adapted ancestrally insecticidal δ-atracotoxins for defensive purposes to protect themselves while searching for females to mate[23]. However, in all studied spider venoms, defensive toxins only represent a small subset of the entire venom cocktail. The vast majority of components causes paralytic or lethal effects facilitating prey capture. Therefore, spiders likely have primarily evolved venom for predation with only a small proportion of their venom components being dedicated towards a defensive role.

A noteworthy exception are the yellow-sac spiders of the genus *Cheiracanthium*, which are well known for causing long-lasting and severe pain[24]. This particularly includes *Cheiracanthium punctorium*, a spider also referred to as Nurse´s thorn finger[25,26]. Its trivial name acknowledges the species unique reproductive biology that heavily revolves around active defense of its eggs[25,26]. This small (~ 15 mm) species, together with its congeners, forms the family Cheiracanthiidae, which are placed in the so-called RTA clade of araneomorph wandering spiders, which account for almost half of all known spider species. When *C. punctorium* reach adulthood and mate, they build egg sacs in high grass, which are inhabited by the parents and aggressively defended against potential threats to safeguard the egg clutch[25,26]. This is initially done by both parents, but the male succumbs shortly after mating and the female continues to defend the offspring until its own death some weeks later[25,26]. This happens at the dawn of the central European winter time and the, by then, freshly hatched juveniles hibernate within the egg sac before they disperse from it in the following spring[25,26]. At the time when the egg sacs are defended, adult specimen stop foraging and instead utilize their venom system for defensive purposes[25,26]. In line with bona fide pharmacology of defensive venoms, and unlike most other spiders, *C. punctorium* bites cause immediate intense pain that lasts several hours, sometimes days, and is accompanied by local effects such as swelling, numbness, and inflammation[27,28]. In rare cases, human victims may require emergency treatment and *C. punctorium* is therefore one of the few Central European spiders considered medically relevant[27]. The effects caused by its venom are referred to as *Cheiracanthium*-syndrome in the medical literature and due to its severely painful bites, some cases of Nurse´s thorn finger mass hysteria occurred in Germany and Austria[29,30].

Chromatographic analysis suggested a unique venom profile of *C. punctorium* and revealed the *C. punctorium* toxins (CPTXs) as major components[31,32]. These are members of the CSTX family, which are small neurotoxins and/or neurotoxicity-enhancing synergistic toxins with four disulfide bonds that usually occur as single-domain peptides[33]. However, among CSTX family peptides, the CPTX form a unique subgroup of double-domain toxins composed of two single CSTX domains connected by a short 16 amino acid linker, which are encoded by an intronless gene[31,32]. They cause cytolytic effects via the formation of helices across cell membranes, and stable and irreversible depolarization of insect muscle fibers, leading to paralysis and death[31,32]. The two domains of CPTXs may be closely related, but further investigation is needed to achieve a detailed understanding of their evolutionary relationship. We also lack information from a sufficiently diverse range of close and more distant relatives, which hinders the analysis of evolutionary trajectories.

Here we present a detailed venom profile of *C. punctorium* and a comparative analysis of venoms from across the spider kingdom, allowing us to investigate the evolution of putatively defensive venom components. The *C. punctorium* venom profile is unique among spiders but similar in structure and composition to venoms with known defensive function. Our data support a complex, multimodal evolutionary history within the CSTX family, giving rise to single-domain and multi-domain toxins in Nurse´s

thorn finger spiders, as well as the convergent recruitment of phospholipase $A_2$ (PLA$_2$). This provides insight into the mechanisms driving the evolution of venoms that fulfil distinct ecological roles.

## Results and discussion
### The unique venom profile of *C. punctorium*

We used a modern venomics workflow based on proteotranscriptomics to characterize the *C. punctorium* toxin repertoire in detail. RNA was extracted from pooled dissected venom glands of 12 adult female and eight adult male spiders and sequenced using Illumina technology. The transcriptome was then assembled using a custom bioinformatics pipeline based on multiple assemblers. Venom collection methods influence the composition of the proteome[34], so we collected crude venom from the same spiders 72 and 144 h before dissection using a mock predator to ensure we induced a defensive response. The collected venom was digested with trypsin, analyzed using an Orbitrap Eclipse mass spectrometer, and assembled onto the *C. punctorium* venom gland transcriptome as a species-restricted database for peptide searches.

We found that *C. punctorium* venom contains 40 polypeptides representing 15 protein families (Fig. 1). The identified components ranged in size from 5.8 to 45.1 kDa. This differs strikingly from spider venoms used for hunting, which often contain hundreds or thousands of small neurotoxic peptides[11,13,35]. One of the unique features of *C. punctorium* venom was that CPTXs accounted for 58.1% of all transcripts and thus form the dominant protein family. So far, ten members were identified, only one of which (CPTX-1c) was already known. Interestingly, most identified CPTX follow the well-known double-domain nature of this subgroup. However, we also identified one-domain members of this family (CPTX-2d, CPTX-5a, and CPTX-5b), yet these only account for a marginal fraction of the overall venom (8.4% of summed-up TPM). Most of the retrieved venom profile is constituted by the double-domain CPTX, which take an outstanding position as they are often extremely potent because their double-domain architecture enables bivalent interactions[16,31,32]. That said, they are rarely found in the animal kingdom, and have only been identified in a single crustacean (*Xibalbanus tulumensis*, Remipedia) and three spiders (*Haplopelma schmidti*, Theraphosidae; *Hadronyche infensa*, Atracidae; and *C. punctorium*, Cheiracanthiidae)[13,16,32,36]. The double-domain toxins account for a small fraction of the overall venom in most of these cases, but were considered the major components of *C. punctorium* venom based on chromatographic peak integration[31,32]. Our proteotranscriptomic analysis supports this hypothesis because double-domain CPTXs accounted for almost 40% of all venom transcripts. *C. punctorium* therefore, appears unique among spiders and other arthropods in terms of the predominance of double-domain proteins. Moreover, facing the high diversity of double-domain toxins recovered, *C. punctorium* venom represents the most promiscuous library of double-domain toxins so far discovered.

The second most abundant group of proteins was protease inhibitors (12.6%), followed by serine proteases (10.2%), venom protein 11 (9.1%), and PLA$_2$ (5.0%). The remaining 10 protein families, including known spider toxin families such as neuropeptide-like components, non-CSTX ICKs, and others, together accounted for only 4.9% of transcripts verified at the proteome level. Overall, 69.8% of all transcripts in our proteotranscriptomic dataset were annotated as putative toxins, 12.7% as putative enzyme inhibitors, and 17.5% as putative enzymes (Fig. 1). Although spider venoms with many more non-neurotoxic components are already known[37,38], *C. punctorium* is so far the only spider that has recruited serine proteases and PLA$_2$ into its venom system as major components.

Our analysis on the *C. punctorium* venom profile reveal insights into the mechanisms at play in manifesting *Cheiracanthium*-syndrome and its painful symptoms when the typical biological activities from the recovered protein families are considered. On one hand, defensive envenoming is often accompanied by cytotoxic effects that lead to localized cell death, swelling, inflammation, and pain: For instance, in cobra venom, the cytotoxic activity is considered an evolutionary adaption to defensive bite that evolved in tandem with hooding behavior[5,39]. In light of this association,

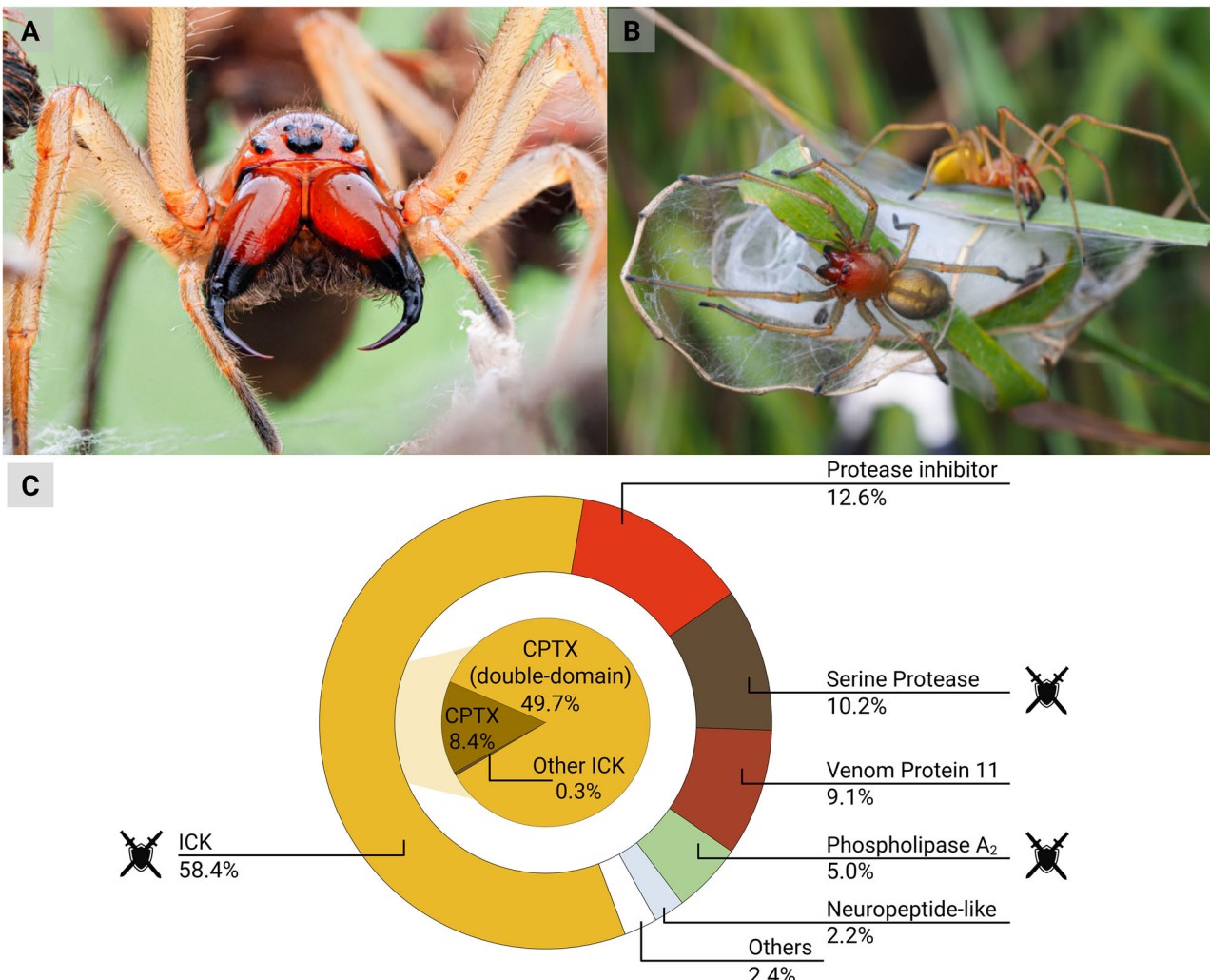

**Fig. 1 | The largely defensive venom of Nurse´s thorn finger spiders (*Cheiracanthium punctorium*). A** Habitus and defensive behavior of *C. punctorium*, displayed by an adult female presenting its opened chelicerae towards a potential threat as a defensive gesture, **B** a male and female on their breeding cocoon, which they defend aggressively when disturbed. **C** The venom profile of *C. punctorium* revealed by proteotranscriptomic analysis and based on abundances (summed expression values for each toxin family relative to the total). Sword and shield icons indicate venom components causing symptoms associated with nociception and hence may facilitate defense. Created in BioRender. Lüddecke, T. (2025) https://BioRender.com/s6xow9w.

several components of the *C. punctorium* venom profile may serve a defensive function by exerting such activities, yet additional experiments are required to determine their bioactivity. The venoms major component are CPTX that have been previously identified from *C. punctorium* venom and that are known to exert cytolytic activities via targeting cellular membranes leading to cell death and insecticidal activity[31,32]. Particularly due to the latter effect, it cannot be ruled out that a specific, hitherto unknown receptor exists that is targeted by CPTX and plays a role in manifesting neurological symptomatics[31,32]. However, their potent cytolytic activity may play a crucial role in manifesting nociception. A similarly important, and putatively defensive component, is present in form of PLA$_2$. These hydrolyze acyl bonds of membrane phospholipids, which leads to the release of lysophospholipids and fatty acids, including arachidonic acid, among others[40]. Not only is this effect often linked to cell death, but the released compounds are key signaling molecules that disturb multiple pathways, including pain signaling[41,42]. Therefore, PLA$_2$ are well-known to cause local inflammation and pain and are therefore considered defensive venom components. Finally, spider venom serine proteases on one hand serve a physiological role in the venom system by processing venom components, but are also believed to target local tissue at the site of envenoming[43]. There, they supposedly serve as spreading factors and are involved in the manifestation of long-term pain and inflammation[43].

In conclusion, the venom profile of *C. punctorium* and the predicted bioactivities of its components are largely congruent with the painful symptoms of defensive envenomation and thus may underpin its function[8]. These symptoms likely stem from the membrane disrupting and neurotoxic effects of CPTXs[1,32], but other components, such as PLA$_2$ or serine proteases, might also contribute to the algogenic symptomatology induced by *C. punctorium* bites[42,43].

**Comparative venomics reveals the origin of CSTXs at the base of araneomorphs**

Our data suggest that the mostly defensive utilization of venom by *C. punctorium* has resulted in unique selection pressure that gave rise to specialized venom chemistry mostly based on a single neurotoxin family and a few enzymes, including PLA$_2$. A large fraction of the venom therefore appears to target cell membranes instead of ion channels and receptors, which are the primary targets of other spider venoms[14].

To test this hypothesis, we carried out a comparative venomics experiment across the two main radiations of araneomorph spiders: the Araneoidea (orb-weavers and their kin) and the RTA clade (wandering spiders). We compared the araneoids *Larinioides sclopetarius* (Araneidae) and *Meta menardi* (Tetragnathidae) with *C. punctorium* and a closely related member of the RTA clade, *Thanatus vulgaris* (Philodromidae)[12,44].

**Fig. 2 | Comparative venomics across the spider kingdom.** The simplified cladogram shows phylogenetic relationships[44] between the spiders we analyzed (photographs on the right) and the donut charts illustrate the composition of each venom measured as relative summed-up TPM. Created in BioRender. Lüddecke, T. (2025) https://BioRender.com/1yq37fv.

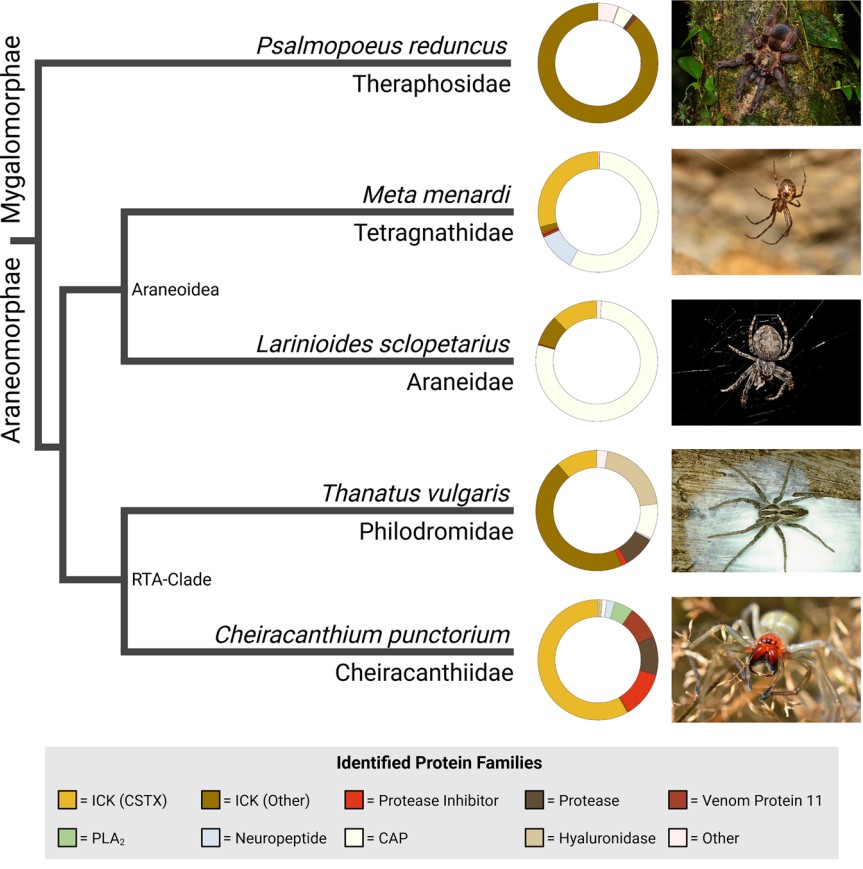

**Identified Protein Families**

■ = ICK (CSTX)  ■ = ICK (Other)  ■ = Protease Inhibitor  ■ = Protease  ■ = Venom Protein 11

■ = PLA₂  ■ = Neuropeptide  □ = CAP  ■ = Hyaluronidase  □ = Other

Venom collection from these small species is challenging[45]. We therefore sequenced the venom gland transcriptomes from several pooled individuals per taxon (Supplementary Table 1). Venom glands were dissected 72 h after feeding to induce replenishment. The resulting venom profiles were annotated, allowing the phylogenetic placement of each source organism to investigate patterns of spider venom evolution (Fig. 2).

Our comparison of these diverse venom profiles shed light on compositional trends between lineages, particularly the distribution of CSTXs. These toxins were found in all four species but contributed differently to each profile. In *M. menardi* and *L. sclopetarius*, they accounted for 29.1% and 12.3% of toxins, respectively. Both species also featured abundant CAP proteins, similar to other predominantly web-hunting spiders[37,38], as well as additional ICK toxins, although more neuropeptide-like components were detected in *M. menardi*. The venom gland transcriptome of *T. vulgaris* featured abundant ICK toxins (56.8%), hyaluronidases (20.3%), CAP proteins (9.5%), proteases (8.6%), and protease inhibitors (1.5%). A considerable fraction of the venom was attributed to the CSTX family (10.9%) and several of the toxins were similar to those found in *C. punctorium*. Given the relatively close relationship between the two species, the sequence similarity suggests that the plesiotypic CSTX that gave rise to the highly specialized apoptypic CPTX in *C. punctorium* was already present in the common ancestor of the araneomorphs.

Although CSTXs were abundant and widespread in these araneomorph venoms, no known representatives have arisen from the more primitive mygalomorphs. The greatest diversity of CSTXs has been isolated from members of the RTA clade[46]. This indicates that CSTXs have acquired a more pivotal role within the venom systems of wandering spiders compared to other, foraging web-building araneomorphs. The complete absence of CSTXs in mygalomorph venoms suggests that these toxins evolved in early araneomorphs. To test this hypothesis, we carried out an additional transcriptomic survey of the venom gland from the Costa Rican orange mouth tarantula (*Psalmopoeus reduncus*), representing the Psalmopoeinae

subfamily of Theraphosidae, the most prominent mygalomorph lineage. This revealed a complex landscape of >600 toxins, similar to other Theraphosidae[47,48]. *P. reduncus* venom features hundreds of neurotoxins, predominantly from the neurotoxin 10 (Huwentoxin-I), neurotoxin 12 (Huwentoxin-II), and neurotoxin 14 (Magi) families, and also contains a few enzymes, including serine proteases and CAP proteins. Although we identified 26 families of neurotoxins in the *P. reduncus* transcriptome, no CSTX transcripts were retrieved. CSTX therefore appear to be evolutionarily restricted to modern araneomorph spiders, suggesting they evolved after the mygalomorph–araneomorph split more than 300 mya[44].

## Phylogenetic analysis suggests a complex, multimodal evolution of CSTX domains

We investigated the available sequence space in more detail to clarify the evolutionary history of single-domain and double-domain toxins in the CSTX family. We retrieved all CSTX sequences from our study and supplemented them with all known gold-level annotated homologs archived in the VenomZone database[46].

A family-wide comparative alignment showed that all known CSTXs outside the Cheiracanthiidae retained a traditional one-domain architecture, but we also found three single-domain toxins in *C. punctorium* (CPTX-2d, CPTX-5a, and CPTX-5b), see Fig. 3. All three examples corresponded to domain 1 of the double-domain CPTX and still comprise the typical CSTX disulfide crosslinking, with the exception of CPTX-5b, which has lost two of its cysteines. The absence of a lone second domain in *C. punctorium* supports the independent evolution of the double domain within the family Cheiracanthiidae, followed by diversification that gave rise to the unique venom components of *C. punctorium*: Spider venom ICK toxins are derived from an ancestral primordial peptide that diversified into multiple families with different cysteine scaffolds and distinct diversification trends and trajectories[13,49]. In some cases, toxin fusion created pharmacologically relevant double-domain toxins, such as those identified in

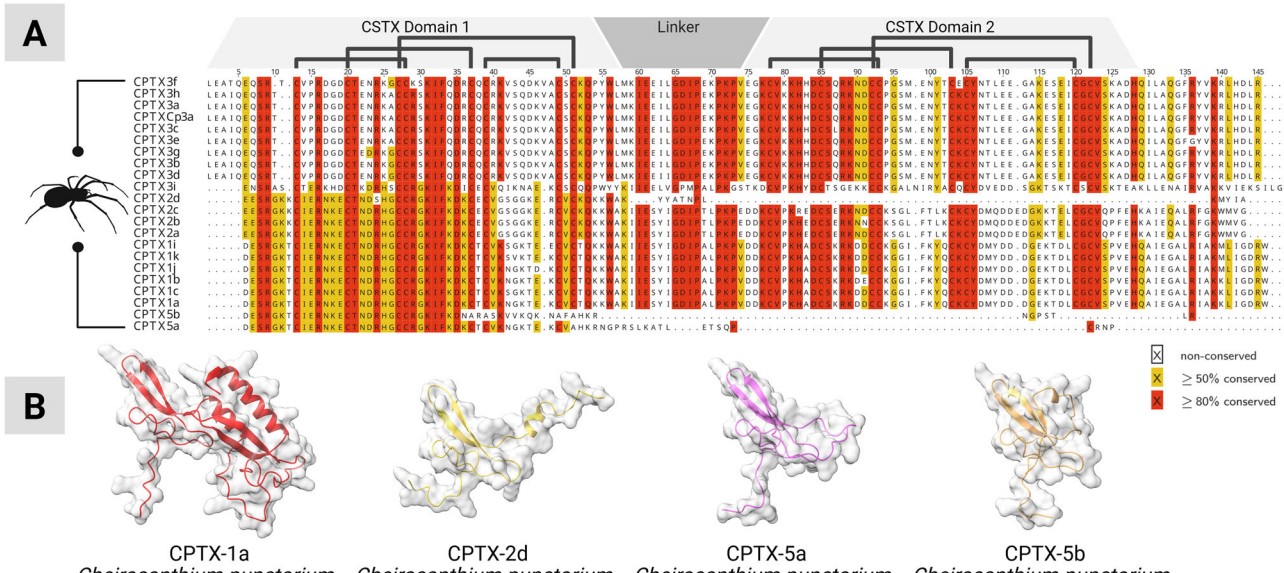

**Fig. 3 | Alignment of CPTX sequences. A** Sequences identified in this study aligned to all previously described homologs. The position of the two CSTX domains and their short linker is highlighted. Connective lines indicate the cysteine crosslinking within each domain. The color code indicates sequence similarity per position in the alignment. **B** Structures of selected double-domain (CPTX-1a) and single-domain (CPTX-2d, CPTX-5a, and CPTX-5b) toxins identified in *C. punctorium* venom as predicted by Alphafold 3. Created in BioRender. Lüddecke, T. (2025) https://BioRender.com/qo7dayd.

*C. punctorium*[13]. To understand this process in more detail, we built a maximum likelihood tree of the entire CSTX family (Fig. 4), including the individual domains of each double-domain toxin. The resulting phylogeny featured 243 peptides from eight spider families, forming at least five distinct clades. The first and largest (group A) features toxins from four families (Tetragnathidae, Araneidae, Lycosidae, and Philodromidae), whereas group B contains toxins from two families (Zodariidae and Philodromidae). Group C contains toxins from four families (Philodromidae, Thomisidae, Zodariidae, and Tetragnathidae), whereas groups D and E feature toxins from all the families except Thomisidae and Zodariidae. The groups were designated as sisters with 86% support. Of particular interest was the placement of each domain of the double-domain CPTXs. All members corresponding to domain 1 (the N-terminal domain) clustered in group D, whereas all members corresponding to domain 2 (the C-terminal domain) clustered in group E. The most common evolutionary mechanism for the creation of double-domain toxins is ancestral gene duplication followed by tandem gene fusion[13,50,51], but in such cases, we would expect the two domains to form a monophyletic clade. The placement of the two domains in separate, well-supported clades (98% for group D and 85% for group E) argues against such a mechanism and instead indicates the independent evolution of each domain prior the fusion event.

Several araneomorphs have recruited multiple CSTX groups into their venom systems[37,43,52,53]. All five groups are present in the family Philodromidae, whereas the family Tetragnathidae features all except group B. The families Araneidae and Lycosidae have recruited all except groups B and C. Three families have only two groups: Zodariidae (groups B and C), Trechaleidae, and Cheiracanthiidae (groups D and E for both). Only one family is restricted to a single CSTX group, namely Thomisidae (group B). Spider venoms in general and CSTXs in particular are known to evolve under strong purifying selection[11,31,32,37]. For metabolically expensive traits, such as venom, it has been shown that such purifying selection causes the loss of spider venom components once they are either rendered superfluous (e.g., due to trophic specialization) or functionally replaced by economically frugal alternatives[54,55]. For instance, this process has caused the loss of venom components in orb-weaver spiders, which switched largely to web-based foraging instead of venom-based hunting[37], and even caused the entire loss of the venom system in Uloboridae, which completely suspended the venom application during hunting[56,57]. With these dynamics in mind, our data support the following scenario underlying the evolution of CSTXs, especially the emergence of double CPTX domains:

Following the initial recruitment of an ancestral CSTX in early araneomorphs, before the split from Araneoidea and the RTA clade, multiple gene duplication events gave rise to the multiple lineages that are present in both major araneomorph clades. Some families retained most or all of their CSTX genetic diversity, whereas others lost part or even most of their portfolio due to purifying selection. This is supported by strong evidence of purifying selection particularly in modern araneomorphs[11,31,58,59]. The double-domain toxins arose from CSTX groups D and E, both of which were already present and undergoing independent evolutionary trajectories in early araneomorphs, through a gene fusion event. This was followed by further gene duplication and divergence to generate the double-domain toxin repertoire we see today. Some of the group D toxins are still retained as single-domain proteins, whereas CPTX domain 2 has become extinct as an independent module, at least in the species we investigated. Comparative genomics emerged as an important element to accurately decipher such evolutionary dynamics within venom systems, but high-quality genomes of spiders are scarce[11]. Hence, at time of writing, analyzing these processes on genomic level is pending. Recently, a genome for *C. punctorium* has been sequenced (NCBI Bio-Project PRJNA1040011). Once a broader assembly of spider genomes, particularly from within the RTA-clade, becomes available, this key resource may pave the way to further disentangle the evolutionary history of CSTX.

### Defensive PLA₂ proteins were convergently recruited into *C. punctorium* venom

The venom profile of *C. punctorium* contains abundant compounds with likely cytolytic activity that may be the active principles behind its pain-causing effects, which is unprecedented among spiders. Several such compounds have been found in small quantities in the venom of other spiders (e.g., CAP proteins or serine proteases)[15], but the surprising abundance of PLA₂ proteins warrants deeper discussion.

PLA₂ is a family of small enzymes (< 20 kDa) often found in venom[60]. They catalyze the hydrolysis of phospholipids and cause a wide range of envenomation symptoms, including local, coagulotoxic, and neurotoxic effects[60]. PLA₂ is an important component of defensive venoms, and is associated with the evolution of proalgesic venom in cobras[5]. PLA₂ has been

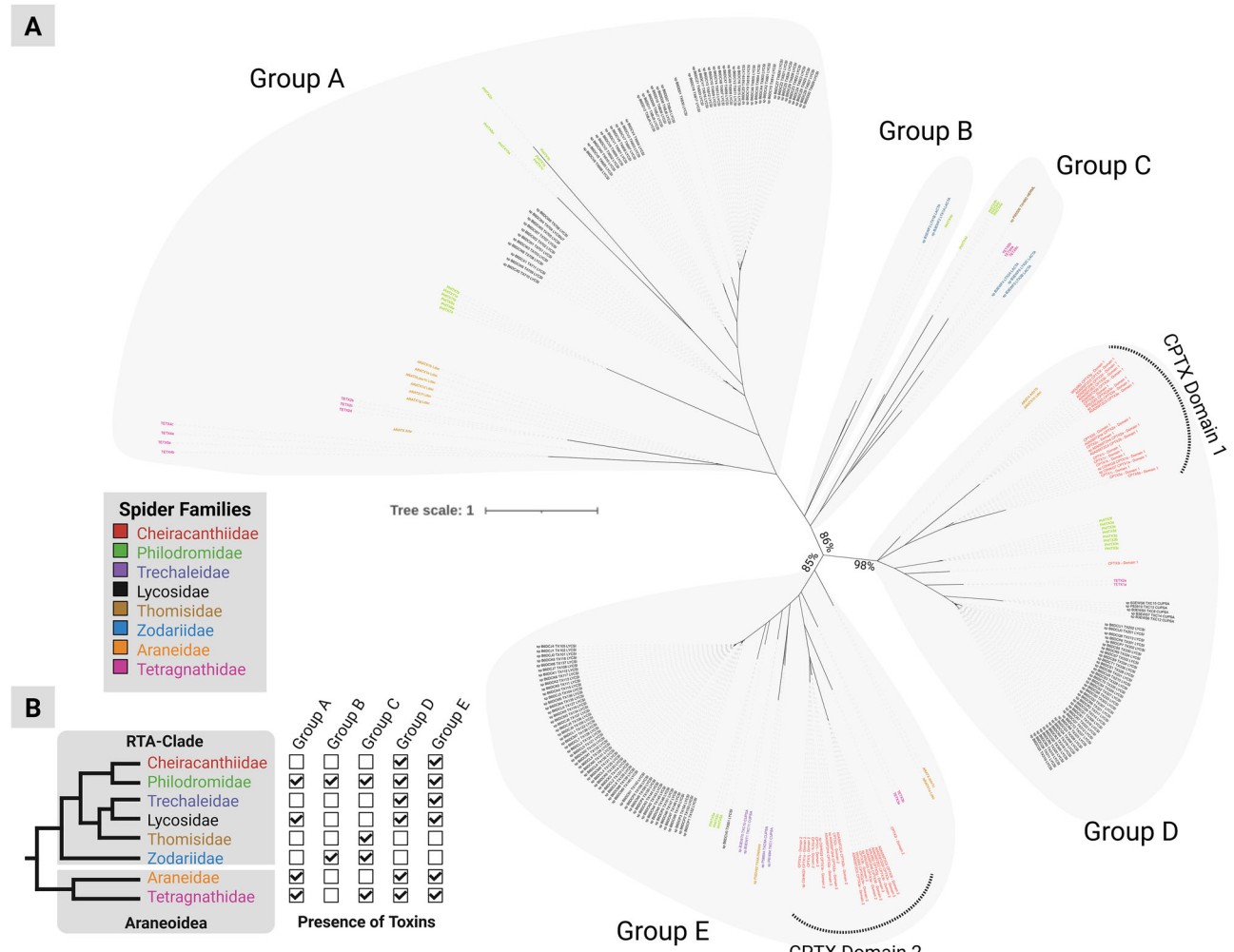

**Fig. 4 | Molecular evolution of CSTXs and their domains. A** Molecular phylogeny of 243 sequences based on a maximum likelihood analysis. The unrooted tree suggests five major lineages of the CSTX family. Grey areas highlight the clades and the color code indicates the taxonomic origin (spider family level) of each sequence. **B** Presence and absence of CSTX lineages across the spider tree of life. The phylogeny of spider families[44] in which CSTXs have been described is used to indicate which major CSTX lineages occur in each family. Created in BioRender. Lüddecke, T. (2025) https://BioRender.com/vj6qv96.

identified in many arthropods with defensive stings, including scorpions, but particularly hymenopterans such as ants and bees[61–63]. Bee venom, which is the textbook example of defensive functionality, is well known for its high $PLA_2$ content and potency[64]. In contrast, $PLA_2$ has only recently been identified in spider venom, and typically only a small number of proteins are present and in tiny amounts[65]. For instance, a single $PLA_2$ was identified in the venom of *Lampona* sp. (out of 208 components; 0.5% of the venom diversity), *Hylyphantes graminicola* (out of 110 components; 0.9% of venom diversity), *Physocyclus mexicanus* (out of 116 components; 0.9% of venom diversity), *Tetragnatha versicolor* (out of 62 components; 1.7% of venom diversity), and *Acanthoscurria juruenicola* (out of 92 venom components; 1.1% of venom diversity respectively[38,66–69]. Additionally, two out of 157 components were identified as $PLA_2$ in the venom of *Stegodyphus mimosarum*, which translates into 1.3% of the species venom diversity[70]. In *Cupiennius salei*, one of the few species that were investigated quantitatively, the abundance of $PLA_2$ within the venom was 0.04%[43]. In agreement with that, several other previously studied spiders did not yield $PLA_2$ at all[60], similar to the taxa that we screened in our comparative venomics experiment, which were devoid of noteworthy $PLA_2$ amounts (Fig. 2). In contrast, *C. punctorium* abounds with a rather high diversity and abundance of those. On the diversity-level, its venom contains 7.5% and on abundance-level 5.0% of $PLA_2$, respectively (Fig. 1). Hence, it contains an approximately seven-times higher known diversity and 125-times higher quantitative amount of $PLA_2$ when compared to previously investigated spider venoms.

This agrees with previous bioactivity profiling studies, which revealed high levels of $PLA_2$ activity in the closely related *C. mildei*[71]. Given the frequent utilization of *C. punctorium* venom for defense and given the well-established role of $PLA_2$ in this context, as well as considering that other spiders primarily use their venom for prey capture, the abundance of $PLA_2$ in *Cheiracanthium* may best be explained by convergence. Animal venoms are well known for their tendency to convergently recruit non-venom proteins and weaponize them into venom components[4]. Usually, a small subset of proteins is weaponized to serve similar functions, with examples including multiple instances of convergent evolution of major toxin families from all across the animal kingdom[2–5,72]. In the case of *C. punctorium* venom, our data suggest that $PLA_2$ proteins have been convergently recruited as major components to serve a defensive function.

Sequence analysis of *C. punctorium* $PLA_2$ proteins revealed they are most similar to homologs in hymenopterans (bees of the genera *Apis* and *Xylocopa*) and scorpions of the families Buthidae, Hemiscorpiidae, and Scorpionidae (Supplementary Data 1). The high level of sequence similarity is particularly evident in the N-terminal portion (Fig. 5). We used Alphafold 3 to build models of the homologous proteins, revealing extensive structural similarities, especially between *C. punctorium* and hymenopteran $PLA_2$ (Fig. 5B). The latter are well known for their pro-inflammatory, cytotoxic, and proalgesic effects[40,64,73–75], and the sequence similarities of *C. punctorium* $PLA_2$ indicate a similar function. The evolutionary convergence of $PLA_2$ sequences and structures in the defensive venoms of arthropods from at

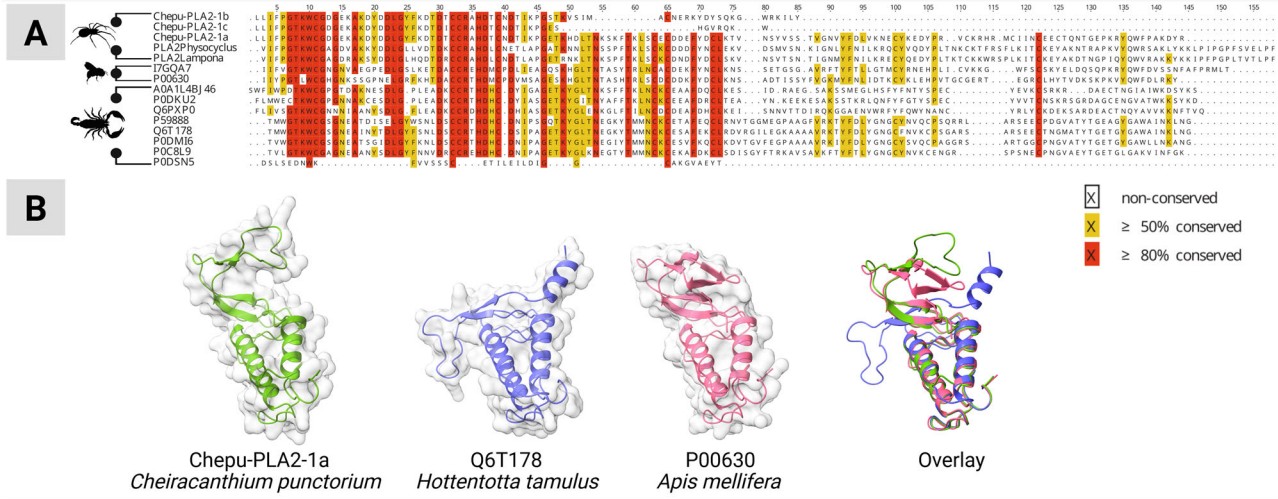

**Fig. 5 | Alignment of *C. punctorium* PLA₂ proteins with homologs in other species. A** Multiple sequence alignment of all *C. punctorium* PLA₂ proteins with homologs in other spider venoms and close relatives from bees and scorpions. **B** Alphafold-predicted structures of selected PLA₂ proteins—from left to right Chepu-PLA2-1a (*C. punctorium*, spiders), Q6T178 (*Hottentotta tamulus*, scorpions), P00630 (*Apis mellifera*, bees)—and an overlay of all three. Created in BioRender. Lüddecke, T. (2025) https://BioRender.com/o4d11la.

least three different taxonomic lineages elegantly shows how nature uses similar molecular solutions to solve related functional problems across diverse animal lineages.

## Conclusion

We have presented a detailed analysis of one of the few predominantly defensive spider venoms. Our proteotranscriptomic profiling of *C. punctorium* revealed a unique venom profile primarily composed of CSTX-family double-domain toxins as well as potentially cytolytic proteins such as PLA₂. Our findings explain the painful reactions triggered by *C. punctorium* bites and provide insight into the medical and ecological relevance of the venom.

While our study provides a snapshot into the largely unstudied area of spider venom utilized beyond purely trophic function, several key questions remain unanswered. On one hand, this particularly includes the bioactivity of identified components. While for double-domain CPTX, some functional data is available, little is known about spider venom PLA₂ and serine proteases[60]. Therefore, we urgently require in depth functional screens to validate our hypotheses. Even for the better studied CPTX, it remains unclear whether their effects are purely based on membranolytic properties or if they interact with a hitherto undiscovered receptor[31,32]. We recommend future studies to employ heterologous expression to produce these venom components to experimentally determine their activities. Likewise, the insecticidal activity of CPTX remains an interesting conundrum. Since adult *C. punctorium* are supposedly not actively hunting and, instead, switch to a sedentary lifestyle to defend their eggs, it is astonishing that their major venom component retained insecticidal activity[31,32]. With this in mind, it is interesting that the defensively employed δ-atracotoxins of atracids similarly retained their insecticidal activity[23]. This raises the question whether these toxins still serve some form of trophic role, if their insecticidal activity is simply a conserved evolutionary leftover, or if a defensive function against insect predators or parasitoids plays a role in this context. Since little literature on the natural history of *C. punctorium* is available, it can not be ruled out that some trophic activity is still present in adult specimens but has been overlooked thus far. To better grasp whether or not this is the case and to better understand the biochemical ecology of the species and its venom, it would be fascinating to explore whether an ontogenetic shift of venom profiles is present and if adults before and after mating feature different venom profiles, which would be indicative of varying functional context. These works should be paired with extensive natural history assessments and ethological studies under laboratory conditions.

Venom components in the infraorder Araneomorphae have been mostly overlooked in favor of larger species that are easier to study and from which large amounts of venom can be extracted[76]. The analysis of more challenging species is necessary to generate a complete picture of venom diversity, especially among the most successful spider lineages[77]. Our proteotranscriptomic analysis enabled us to build a pan-infraorder comparative phylogeny of spider venom profiles and thereby revealed several fascinating aspects of CSTX family evolution. We pinpointed the evolutionary origin of this family (coinciding with the mygalomorph-araneomorph split) and identified clades and divergent lineages that shed light on its radiation. This led to a hypothesis on the complex, multimodal evolutionary history of the double-domain CPTX toxins. It aligns with recently proposed evolutionary models for spider ICK toxins more broadly, where molecular diversity appears to stem from a single primordial structure that subsequently gave rise to an expanded arsenal of weapons[13,49].

The most striking revelation was the remarkable degree of sequence and structural similarity between PLA₂ proteins in *C. punctorium* and hymenopterans, which shows how venom systems can convergently weaponize similar plesiotypic proteins to address related challenges across different animal lineages.

## Material and methods
### Collection and rearing of spiders

Araneomorph spiders (*C. punctorium*, *T. vulgaris*, *M. menardi*, and *L. sclopetarius*) were collected in Germany (see Supplementary Table 1 for collection details). The mygalomorph *P. reduncus* was sourced from the pet trade. Before sampling, spiders were kept at room temperature with a ~ 12-h photoperiod for at least 1 week in appropriately-sized plastic enclosures (*P. reduncus* = 30 × 30 × 30 cm, all others 10 × 10 × 10 cm) containing potting soil as a substrate.

### Venom collection

The strategy used to collect venom was recently shown to affect venom composition, so we modified the mock-target system to trigger defensive bites from *C. punctorium*. Briefly, the openings of 1.5-ml Eppendorf tubes were covered with thin-drawn Parafilm squares (1 × 1 cm) and gripped with dissection forceps. Spiders were provoked with the Parafilm-covered side of the tube until they deployed a defensive bite. Small droplets of venom near visible depressions in the film were collected using a 10-μl pipette tip. Venom milking was carried out 144 and 72 h before venom gland dissection. This approach was selected because venom glands from wild-caught

spiders are exhibiting variable physiological states with distinct RNA and protein landscapes. It has thus been declared best practice to apply such pre-milkings at selected time points to normalize the glands expression landscape due to forced venom replenishment on the linked onset of transcription. These pre-milkings should be performed several days prior the dissection and, based on our own experience, intervals of 48 h or 72 h are best suited for spider venom glands. The collected venom samples were pooled, lyophilized, and stored at −80 °C.

### Collection of venom glands, RNA isolation, and sequencing

Venom gland replenishment was triggered in *C. punctorium* by venom collection and in the other species by feeding with a small cricket 72 h before venom gland dissection, thus ensuring all venom glands were at equivalent stages of gene expression/protein synthesis. All glands were dissected from $CO_2$-anesthetized specimens under a Zeiss Stemi 508 stereomicroscope, washed in distilled water, and transferred to 1-ml RNAlater solution (Sigma-Aldrich). The glands from members of the same species were pooled and stored at −80 °C.

RNA extraction and sequencing were outsourced to Macrogen. Following RNA extraction, libraries were constructed using the Illumina TruSeq RNA Sample Prep Kit v2 or TruSeq Stranded mRNA Library Prep Kit (paired-end, 151-bp read length). Quality was controlled by the verification of PCR-enriched fragment sizes using an Agilent Technologies 2100 Bioanalyzer with a DNA 1000 chip. The library quantity was determined by qPCR using the rapid library standard quantification solution and calculator (Roche). The libraries were sequenced using Illumina technologies, further details are provided in Supplementary Table 1. Raw sequencing data are available at the European Nucleotide Archive (Study PRJEB86488).

### Transcriptome sequencing and analysis

Transcriptome data were processed using a modified version of our in-house assembly and annotation pipeline[62]. Details of the software and database versions used for each sample are provided in Supplementary Table 1. All input sequences were inspected using FastQC v0.11.9/0.12.1 (www.bioinformatics.babraham.ac.uk) before trimming with cutadapt v4.2/4.9[78] using a quality cutoff of 28 and a minimum length of 25 bp or 30 bp. The trimmed reads were corrected using Rcorrector v1.0.5/1.0.7[79] and assembled de novo using a pipeline incorporating Trinity v2.13.2/2.15.1[80] with a minimum contig size of 30 bp and maximum read normalization of 50 and rnaSPAdes v3.15.5[81] with and without error corrected reads. Both rnaSPAdes assemblies were built with k-mer sizes of 21, 33, and 55. If strand specificity was available, this was taken into account during assembly with the respective parameters. All contigs were combined into a single assembly, in which transcripts from all assemblers were merged with either CD-HIT-EST v4.8.1[82] or fastanrdb v2.4.0 (github.com/nathanweeks/exonerate) if they were identical. The reads were remapped to the assembly using HISAT2 v2.2.1[83] and expression values (transcripts per million, TPM) were calculated using StringTie v2.2.1/2.2.2[84]. SAM and BAM files were converted using samtools v1.16.1/1.20[85]. Open reading frames (ORFs) were then predicted with TransDecoder v5.5.0/5.7.1 (github.com/TransDecoder/TransDecoder) with a minimum length of 10 amino acids and provided for proteome analysis.

### Proteomics

Our bottom-up proteomics strategy involved a mass spectrometry (MS) protocol already applied to other animal venoms[86]. Briefly, we dissolved 10 μg of sample material in 25 mM ammonium bicarbonate containing 0.6 nM ProteasMax (Promega). We added 5 mM DTT and incubated for 30 min at 50 °C to complete the disulfide reduction, followed by modification with 10 mM iodacetamide for 30 min at 24 °C. After quenching the reaction with excess cysteine, we added a 50:1 ratio of trypsin and digested the venom for 16 h at 37 °C. After stopping the reaction by adding 1% trifluoroacetic acid, we purified the sample using a C18-ZipTip (Merck-Millipore), dried it under vacuum, and redissolved the material in 10 μl 0.1% trifluoroacetic acid.

We separated the peptides on an UltiMate 3000RSLCnano device (Thermo Fisher Scientific), then injected 1 μg of the sample material into a 50 cm μPAC C18 column (Pharma Fluidics, Thermo Fisher Scientific) in 0.1% formic acid at 35 °C. The peptides were eluted in a linear gradient of 3–44% acetonitrile over 240 min. The column was then washed at a flow rate of 300 nl/min with 72% acetonitrile. The separated peptides were injected into an Orbitrap Eclipse Tribrid MS (Thermo Fisher Scientific) in positive ionization mode with the spray voltage set to 1.5 kV and a source temperature of 250 °C achieved using a TriVersa NanoMate (Advion BioSciences). We scanned the samples in data-independent acquisition mode with the following settings: scanning time = 3 s, *m/z* range = 375–1500, and resolution = 120,000. Auto-gain control was set to standard with a maximum injection time of 50 ms. The most intense ions in each cycle with a threshold ion count > 50,000 and charge states of 2–7 were selected with an isolation window of 1.6 *m/z* for higher-energy collisional dissociation (normalized collision energy = 30%). Fragment ion spectra were acquired in the linear ion trap with a rapid scan rate and normal mass range. The maximum injection time was set to 100 ms and selected precursor ions were excluded for 15 s post-fragmentation.

We used Xcalibur v4.3.73.11 and Proteome Discoverer v2.4.0.305 (both from Thermo Fisher Scientific) for data acquisition and analysis. Proteins were identified using Mascot v2.6.2 by searching against the predicted ORFs from our transcriptome analysis using the following settings: precursor ion mass tolerance = 10 ppm, carbamidomethylation as a global modification, methionine oxidation as a variable modification, and one missed cleavage allowed. Fragment ion mass tolerance in the linear ion trap for $MS^2$ detection was set to 0.8 Da and the false discovery rate was limited to 0.01 using a decoy database. For qualitative analysis, we only considered proteins that were identified with a Mascot score of at least 30 and at least two verified peptides. A comprehensive list of all venom components identified with confidence, and their characteristics and annotations, are provided in Supplementary Data 1. Proteomic raw data have been uploaded to ProteomeXchange with identifier PXD061529.

### In silico analysis, structural modeling, and phylogenetic inference

Identified toxin precursors were annotated using InterProScan v5.61-93.0/5.69-101.0[87] and a DIAMOND v2.0.15/2.1.9[88] blastp search against the public available databases VenomZone, UniProtKB/Swiss-Prot Tox-Prot, UniProtKB/Swiss-Prot, and UniProtKB/TrEMBL v2022_05/2024_04 was performed. The *E*-value was set to a maximum of $1 \times 10^{-3}$ in ultra-sensitive mode with all target sequences reported (−max-target-seqs 0). We then calculated the coverage of query and subject, and the similarity with the BLOSUM62 matrix using BioPython v1.81/1.83[89] for each hit. Sorting by similarity, query, and subject coverage for each toxin candidate led to the resulting hit for the final analysis. Precursors without a predicted signal peptide by SignalP v6.0 g/h[90] in slow–sequential mode for eukarya were removed from our dataset. Annotated precursors were aligned to known venom components from the same putative family to verify our assignments using ClustalW in Geneious v10.2.6 (www.geneious.com). For multi-domain CSTX proteins, we annotated distinct domains following alignment to established domains in Screpyard[91], a manually curated database of cysteine-rich peptides. The resulting annotated peptide sequences are provided in Supplementary Data 1–6.

Alphafold 3 in the Galaxy platform was used to generate 3D models of the proteins[92]. Sequences were submitted online and structures were predicted using default settings. The resulting model was downloaded and visualized using ChimeraX[93], retrieved models are provided in the Dryad database[94].

For phylogenetic analysis, CSTX full precursor sequences and/or single-domain sequences of double-domain toxins were aligned using ClustalW in Geneious v10.2.6 and uploaded to the IQ-TREE webserver[95]. A maximum likelihood tree was calculated using protein as the sequence type and the best-fitting substitution model was selected in Auto mode. Branch analysis was achieved by ultrafast bootstrap support using 1000 replicates

and an activated SH-aLRT branch test. We selected a perturbation strength of 0.5, and the IQ-TREE stopping rule was set to 100. From the resulting tree space, a 65% majority rule consensus tree was created (provided as Supplementary Data 6) and visualized using iTOL[96]. Given the absence of protein outgroups, we refrained from rooting the tree and analysed an unrooted configuration. Alignments were visualized with TEXshade[97] in identical mode (Figs. 4 and 5).

## Reporting summary
Further information on research design is available in the Nature Portfolio Reporting Summary linked to this article.

## Data availability

Raw proteomic data are available via ProteomeXchange with identifier PXD061529. Raw transcriptomic data have been uploaded to the European Nucleotide Archive (ENA) (Study PRJEB86488). The following data are available as supplements: Information on sampled spiders and their sequencing (Supplementary Table 1), proteotranscriptomic analysis of *C. punctorium* (Supplementary Data 1), Transcriptomic analysis of *M. menardi* (Supplementary Data 2), *P. reduncus* (Supplementary Data 3), *L. sclopetarius* (Supplementary Data 4), *T. vulgaris* (Supplementary Data 5), as well as the retrieved optimal phylogenetic tree (Supplementary Data 6). Structural models generated in Alphafold 3 are available online in the Dryad database (https://doi.org/10.5061/dryad.fn2z34v7t). All other data are available from the corresponding author upon reasonable request.

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

## Acknowledgements

The authors thank members of the German Arachnological Society for supporting this study. We particularly thank Vanessa Oehmig for providing the spiders used in this study. The authors thank Louis Roth, Vanessa Oehmig, and Tobias Hauke for providing spider pictures used to assemble Figs. 1 and 2. We acknowledge technical assistance of the Bioinformatics Core Facility at the professorship of Systems Biology at JLU Giessen and the provision of IT resources and general support by de.NBI/ELIXIR-DE (W-de.NBI-010) funded by the Federal Ministry of Education and Research. We thank Richard M. Twyman for language editing the manuscript. This work was supported by generous funding from the Hessian Ministry of Science and Art (HMWK) via the LOEWE Centre for Translational Biodiversity Genomics, granted to A.V. V.H. was funded by the Australian Research Council (FT190100482).

## Author contributions

Tim Lüddecke, Volker Herzig, and Andreas Vilcinskas conceptualized the study. Tim Lüddecke further carried out the laboratory work. Fieldwork was conducted by Tim Lüddecke, Thomas Lübcke, and Volker von Wirth. Sabine Hurka, Josephine Dresler, and Volker Herzig generated the transcriptomic data and conducted the data visualization. Sabine Hurka and Josephine Dresler further carried out other bioinformatic analyses, while Tim Lüddecke, Günter Lochnit, and Thomas Timm generated and analysed the proteomic data. Andreas Vilcinskas provided funding. Tim Lüddecke, Josephine Dresler, and Volker Herzig wrote the original draft, all other authors were involved in reviewing and editing.

## Funding

This project was funded by Hisayo Ito from the Winners of the 2017 JA Ōmura Awards for excellence. Open Access funding enabled and organized by Projekt DEAL.

## Competing interests

The authors declare no competing interests.
