## [Transparent Peer review file · Communications Biology]

Comparative venomomics suggests an evolutionary adaptation of spider venom from predation to defense

Corresponding Author: Dr Tim Lüddecke

This manuscript has been previously submitted at another journal. This document only contains information relating to versions considered at Communications Biology.

Version 0:

Reviewer comments:

Reviewer #1

(Remarks to the Author)

Convergent evolutionary adaptation of spider venom from predation to defense

This manuscript describes the venom composition of *C. puncturium* using pooled venom gland transcriptomics and venom proteomics. The authors describe the main venom component as a double domain toxin (CpTx) that is consistent with prior proteomic analyses. The authors then compare the venom composition of *C. puncturium* to four other spiders from different families and infraorders to describe the evolutionary history of the main double domain toxin. Lastly the authors predict the structure of PLA2s in *C. puncturium* and compare to PLA2s in other non spider lineages. The results the authors present are robust and display the composition and potential evolutionary history of *C. puncturium* venom which would be of interest to others within the community. I appreciate the comparative aspect of the paper in determining the evolutionary history of the venom and the focus on describing the main venom component across different spider lineages. I

Major Points:

- 1) My first major comment is the focus and claims on the defensive nature of this venom. The data presented in this paper do not test the functionality of the venom or the behavior of this species. The authors make large and unsupported claims that need to be cleaned up and addressed.
 - In the introduction, the authors describe the life history of *C. puncturium* using a single non peer reviewed source (citation 21). The source does not support the author's claim that *C. puncturium* stop foraging only that they are extremely defensive of the nest. The source also states that only the females guard the nests. Since the claim of this manuscript is that this venom is being primarily used a defensive venom, I would like peer reviewed articles that support the main basis of the manuscript.
 - The authors do include two case studies involving envenomation from *C. puncturium* that from a human perspective make it seem as primarily a defensive venom. However, in the next paragraph describing the toxins, the article they cite that did the functional characterization of the double domain CPTX stated that the venom showed "high insecticidal, cytotoxic, and membrane-damaging activities." Can the authors add additional context to their claim on the defensive nature of this venom when the functional tests show not only defensive mechanisms of action?
 - Figure 1 shows icons next to components with defensive characteristics but there are no citations in the main text describing these criteria. The authors talk about the relationship to PLA2 from other venomous organisms but do not talk about how serine proteases, ICK, or venom protein 11 toxins/enzymes are primarily defensive. I would like to see more discussion on the hypothesized effects of these venom components.
 - Section 2.2: The authors state in the first paragraph that their data supports the defensive nature of the venom when the authors only described the components of the venom and did not test the function of the venom. The next sentence states that therefore a large fraction of the venom acts on cell membranes and not on ion channels. This is not their data and should be put into more context on how previous work described CPTX as primarily working on cellular membranes but as citation 25 points out they cannot rule out the existence of a specific receptor and that more work is needed to determine the target.
 - Section 2.4: The authors state "Given the predominant utilization of *C. puncturium* venom for defense, and considering that other spiders primarily use their venom for prey capture, the abundance of PLA2 may best be explained by convergence." This is another area where the authors are making the claim that this venom is defensive. While the work aligning PLA2 does support some conserved function, the function of PLA2s in spider venom has not been described (citation 45).

Given the different areas outlined above, I would suggest that the authors either provide more citations backing their claims or tone down the defensive claims of the venom and focus more on the evolution of this venom component and potential mechanisms of function.

2) There are a few acronyms that are confusing throughout the paper, mainly CSTX, CPTX, and dICK. Does CPTX refer to all of the toxins in *C. puncturium* or just the two domain containing toxins? Are dICKs the same as double-domain toxins in the CSTX family? Are CSTX family domains ICKs? Do the authors refer to these interchangeably throughout the paper? Can the author clarify and clear up some of the language in the manuscript? I will highlight a few cases where the reader might be confused.

- “Within these, CPTX form a unique subgroup as dICKs containing two CSTX domains that are divided through a short linker sequence of 16 amino acids and are encoded by an intronless gene.”
- “One of the unique features of *C. puncturium* venom was that CPTXs, a single group of CSTX, accounted for 58.1% of all transcripts and thus form the dominant protein family.”
- “Chromatographic analysis suggested a unique venom profile of *C. puncturium* and revealed the *C. puncturium* toxins (CPTXs), dICK toxins from the neurotoxin 19 or CSTX family, as major components”
- “most of the retrieved venom profile is constituted by the dICK CPTX”

3) There is a full section describing CPTXs and PLA2s but not the other major venom components. Can the authors add an additional section describing these venom components and their function? This would add additional support for the function of the venom and provide necessary information to understand the selective pressures placed on the venom as a whole and not just the CPTXs.

4) The last paragraph of the conclusion states “the most striking revelation was the difference between the targeting cell membranes to trigger rather than other spiders which target ion channels.” This was not described in this paper and the data generated in this paper do not support that claim. The “mode of action” and function of this venom was not studied and was not discussed in any detail in this manuscript. Can the authors remove these claims from the conclusion or heavily modify them and provide more context for these claims? The last sentence of this paragraph focuses more on the structure which was described in this manuscript. The authors should shift the focus on this paragraph to their own results.

Minor Points:

- Fourth Paragraph of the intro “The functionally most derived spider..” Needs a citation to back up claim. Could the authors also expand on how this venom is the “most derived”?
- “The two CSTX domains of CPTXs may be closely related, but further investigation is needed to achieve a detailed understanding of their ecological role” I am unsure how looking at the relationship of these two domains can help understand their ecological role? Can the authors please rephrase this statement or expand on it?
- Last paragraph of the introduction: The authors talk about the structure and composition of *C. puncturium* venom in relation to hymenopterans without a citation. This information needs to be supported by citations or moved into the discussion where the authors expand further.
- Some of the percentages in the second paragraph of 2.1 do not align with the pie graph. For example, 40% of all venom transcripts are double domain CPTXs. However, the pie graph shows 58.1% were CPTXs with 8.4% of those are supposed to be single domain CPTXs. Another place is 12.7% are putative enzyme inhibitors, yet the graph shows 12.6%. Can the authors verify the values in the graph and in the text?
- Figure 1: Can you split the insert in the middle to show single and double domain CPTXs? This might help to clarify some of the percentages shown above.
- The abbreviation uCNTX is used without being described in full.
- Last paragraph of section 2.1: The authors state that the venom profile of *C. puncturium* has predicted activities that are congruent with the painful symptoms. There are no citations to back this claim or the claim in the last sentence of section 2.1. Can the authors add more detail about these predicted activities with appropriate citations?
- Second paragraph in section 2.2. Why only test your hypothesis using CPTX and PLA2s? Why not use serine proteases and Venom Protein 11 as well? Can the authors provide a statement for why they focus on these two groups and not the others even though they are more abundant than the PLA2?
- Figure 5: Can the authors use different colors for the three proteins? The different shades of red are hard to see differences between and it would be helpful to see how they line up in the overlay with more distinct colors.

Reviewer #2

(Remarks to the Author)

Lüddecke and colleagues present a comprehensive characterization of the venom composition of the yellow sac spider *Cheiracanthium puncturium*. Notably, this is the first report of a relatively high abundance of phospholipase A2 (PLA2) in a spider venom. The authors also demonstrate that the species' double-domain neurotoxins (CSTXs or CpTxS) originated from a gene fusion event between two distinct single-domain toxin precursors, offering novel insight into the evolutionary mechanisms underpinning venom diversification.

These findings reveal a case of molecular convergence in venom systems across distantly related venomous animals (e.g., spiders, snakes, bees), suggesting that similar ecological pressures can lead to the recruitment of analogous molecular

components across taxa.

After careful evaluation, we believe the manuscript is of high quality and merits publication. However, a number of points require clarification and refinement. Our specific suggestions are outlined below:

Major Comments

Functional classification of spider venoms:

Personally, I am not fully convinced by the binary classification of spider venoms into primarily “predatory” or “defensive” types. At least based on the evidence provided in this manuscript, I find it difficult to fully understand or accept this distinction. The authors argue that most spiders produce predatory venom, in part based on the observation that spider venoms generally do not cause significant harm to predators. However, if *C. puncturium* venom is to be classified as primarily defensive, does this imply that its venom has lost or diminished its capacity to paralyze prey? This assumption is not directly supported. Therefore, I tend to favor an alternative interpretation—namely, that spider venom primarily evolved for predation, but in some cases (such as *C. puncturium*), certain components have been secondarily enhanced to serve a defensive function. Unless the authors can provide more compelling evidence to support a redefinition of venom functionality, I suggest that expressions such as “defensive venoms” and “defensive spider venom” be used more cautiously or softened throughout the manuscript.

Convergent evolution and dICK toxins:

The manuscript emphasizes convergent evolution of spider venom with that of other taxa in the title, particularly in the context of defensive function. However, a portion of the manuscript is devoted to the discussion of dICK toxins, without clearly establishing their contribution to pain induction or identifying which venom components from other lineages they might be functionally analogous to. For readers lacking a strong background in toxinology, this may obscure the connection between dICK toxins and the proposed convergent evolution. If the link between dICK toxins and defensive convergence is limited, the title or main text could benefit from minor revision to improve clarity for a broader readership.

PLA2 abundance as a “major component”:

The authors assert that *C. puncturium* venom is the only known spider venom where PLA2 constitutes a major component. However, given that PLA2 has been identified in other spider venoms, the manuscript should cite or list comparative data from those species to establish a clear quantitative benchmark. This would help define the threshold at which PLA2 can be considered a “major component” in a meaningful context.

Minor Comments

Genomic localization of toxins:

The genome of *C. puncturium* has been published (GCA_038373885.1), albeit in a draft state. It would greatly strengthen the manuscript if the authors could determine the genomic loci of the identified toxin genes. This would add valuable genomic context to their evolutionary interpretations.

Venom gland depletion timing:

The methods state that venom glands were emptied at 144 h and 72 h before sampling. What was the rationale behind selecting these specific time intervals? Clarification of the biological or experimental considerations would be helpful.

Duplicated CPTX2a label in Figure 3A:

There appear to be two entries labeled CPTX2a in Figure 3A. Please verify whether this is a labeling error.

CPTX14a–c categorization and Figure 1B consistency:

CPTX14a–c are classified as PLA2s in Figure 5A, but in Figure 1B, CPTX entries are listed separately and in parallel with PLA2s. Could this be a labeling inconsistency or error in nomenclature? Please double-check.

Color distinction in Figure 5B:

The color scheme used in Figure 5B results in models that are visually indistinguishable in the overlay image. Please consider using a more contrasting color palette to enhance clarity.

Clarification on inference of independent evolution in Section 2.3:

In Section 2.3, the authors state: “The absence of a lone second domain in *C. puncturium* supports the independent evolution of the double domain within the family Cheiracanthiidae,…” Could the authors clarify how the absence of a standalone second domain supports the hypothesis of independent evolution?

On the interpretation of purifying selection and venom system loss (Section 2.3):

In the third paragraph of section 2.3, the authors state:

“Spider venoms in general and CSTXs in particular are known to evolve under strong purifying selection, which regularly leads to the loss of complexity and even entire venom systems.”

I find this statement somewhat confusing. It is unclear to me how purifying selection—typically associated with the conservation of functional elements—could lead to the complete loss of an entire venom system. Perhaps I have misunderstood the intended meaning, but this point could benefit from further clarification in the manuscript.

Version 1:

Reviewer comments:

Reviewer #1

(Remarks to the Author)

The revised manuscript thoroughly addressed my initial concerns and provided additional ecological/ natural history notes to support the majority of their claims. The language and tone of the paper shifted and supported the findings of the authors. I appreciate the thoroughness of the authors in addressing the concerns of both reviewers and I think that the revised manuscript has greatly improved. I do not have any additional comments for the authors.

Reviewer #2

(Remarks to the Author)

I have carefully reviewed the authors' rebuttal and the revised manuscript. The authors have addressed my previous comments in a satisfactory manner, and I have no further comments on the manuscript.

Reviewers' comments:

Reviewer #1 (Remarks to the Author):

Convergent evolutionary adaptation of spider venom from predation to defense This manuscript describes the venom composition of *C. punctatorium* using pooled venom gland transcriptomics and venom proteomics. The authors describe the main venom component as a double domain toxin (CpTx) that is consistent with prior proteomic analyses. The authors then compare the venom composition of *C. punctatorium* to four other spiders from different families and infraorders to describe the evolutionary history of the main double domain toxin. Lastly the authors predict the structure of PLA2s in *C. punctatorium* and compare to PLA2s in other non spider lineages. The results the authors present are robust and display the composition and potential evolutionary history of *C. punctatorium* venom which would be of interest to others within the community. I appreciate the comparative aspect of the paper in determining the evolutionary history of the venom and the focus on describing the main venom component across different spider lineages.

Reply: We thank the reviewer for this critical but constructive evaluation and overall positive feedback. We believe that their comments and concerns increased the quality of our manuscript and we happily included all of them to the best of our abilities. Please find below a point-by-point reply to each of the issues raised by the reviewer.

Major Points:

1) My first major comment is the focus and claims on the defensive nature of this venom. The data presented in this paper do not test the functionality of the venom or the behavior of this species. The authors make large and unsupported claims that need to be cleaned up and addressed.

- In the introduction, the authors describe the life history of *C. punctatorium* using a single non peer reviewed source (citation 21). The source does not support the author's claim that *C. punctatorium* stop foraging only that they are extremely defensive of the nest. The source also states that only the females guard the nests. Since the claim of this manuscript is that this venom is being primarily used as a defensive venom, I would like peer reviewed articles that support the main basis of the manuscript.

- The authors do include two case studies involving envenomation from *C. punctatorium* that from a human perspective make it seem as primarily a defensive venom. However, in the next paragraph describing the toxins, the article they cite that did the functional characterization of the double domain CPTX stated that the venom showed "high insecticidal, cytotoxic, and membrane-damaging activities." Can the authors add additional context to their claim on the defensive nature of this venom when the functional tests show not only defensive mechanisms of action?

- Figure 1 shows icons next to components with defensive characteristics but there are no citations in the main text describing these criteria. The authors talk about the relationship to PLA2 from other venomous organisms but do not talk about how serine proteases, ICK, or venom protein 11 toxins/enzymes are primarily defensive. I would like to see more discussion on the hypothesized effects of these venom components.

- Section 2.2: The authors state in the first paragraph that their data supports the defensive nature of the venom when the authors only described the components of the venom and did not test the function of the venom. The next sentence states that therefore a large fraction of the venom acts on cell membranes and not on ion channels. This is not their data and should be put into more context on how previous work described CPTX as primarily working on cellular membranes but as citation 25 points out they cannot rule out the existence of a specific receptor and that more work is needed to determine the target.

- Section 2.4: The authors state "Given the predominant utilization of *C. punctatorium* venom for defense, and considering that other spiders primarily use their venom for prey capture, the abundance of PLA2 may best be explained by convergence." This is another area where the authors are making the claim that this venom is defensive. While the work aligning PLA2 does support some conserved function, the function of PLA2s in spider venom has not been described (citation 45).

Given the different areas outlined above, I would suggest that the authors either provide more citations backing their claims or tone down the defensive claims of the venom and focus more on the evolution of this venom component and potential mechanisms of function.

Reply: We thank the reviewer for this careful and very detailed examination of our work and for suggesting these important changes, which we've happily included. Overall, we have toned down the language across the entire manuscript to acknowledge the absence of functional data for several venom components and hence the defensive function. Now, we devote parts of our conclusion on the importance to carry out exactly these experiments (Line 399-422). We have further expanded more on the activity spectrum from the major venom components and on why they are likely involved in defensive context and also added a large amount of citations to underpin our assessment (Line 399-422). We have also expanded upon the insecticidal activity of CPTX and discuss this conundrum as a particularly interesting future avenue in the later stages of the manuscript and provided more literature citations on the painful effects caused by *C. puncturium*, and included notes on the potential possibility of a so far undiscovered receptor for CPTX toxins (Line 399-422).

We particularly emphasized more the natural history of the species (Line 81 – 102). Unfortunately, only few studies are available on the natural history of spiders, especially those from Central Europe and the scarce literature available is usually published in small bulletins or the grey-literature and in German language only. In this case, we have added the best available source published in the peer-reviewed bulletin of the German arachnological society (*Arachne*) which reports natural history observations made until 2023 in context of *C. puncturium*'s denomination to the European spider of the year (see references). Since this work is difficult to find or access online, we also forward a pdf version of the paper to the handling editor to make it available to the reviewers. The expansion on *C. puncturium* natural history is important because it sets the framework for the functional context. In this species, like in most central European spiders, individuals have a one-year lifespan that climaxes in reproduction and oviposition. In case of *C. puncturium*, male and female mate and deposit eggs that are jointly defended. Hunting does not occur anymore at this stage. However, the males die significantly earlier than the females, hence after initial jointly inhabiting the nests, at later stages the defense is facilitated by the surviving female until it also succumbs (hence the misconception that only females safeguard the offspring). After that, the juveniles hatch and hibernate in the egg sac, hatch from it in next spring and then disperse. We have described this now in more detail to clarify the life cycle of *C. puncturium* to the reader. In order to further underscore the matter, we have rephrased the trivial name of *C. puncturium* across our manuscript. While it is often referred to as yellow-sac spider, in Central Europe they are also known as Nurse's thorn finger because of their defensive behaviour safeguarding their young.

2) There are a few acronyms that are confusing throughout the paper, mainly CSTX, CPTX, and dICK. Does CPTX refer to all of the toxins in *C. puncturium* or just the two domain containing toxins? Are dICKs the same as double-domain toxins in the CSTX family? Are CSTX family domains ICKs? Do the authors refer to these interchangeably throughout the paper? Can the author clarify and clear up some of the language in the manuscript? I will highlight a few cases where the reader might be confused.

- "Within these, CPTX form a unique subgroup as dICKs containing two CSTX domains that are divided through a short linker sequence of 16 amino acids and are encoded by an intronless gene."
- "One of the unique features of *C. puncturium* venom was that CPTXs, a single group of CSTX, accounted for 58.1% of all transcripts and thus form the dominant protein family."
- "Chromatographic analysis suggested a unique venom profile of *C. puncturium* and revealed the *C. puncturium* toxins (CPTXs), dICK toxins from the neurotoxin 19 or CSTX family, as major components"
- "most of the retrieved venom profile is constituted by the dICK CPTX"

Reply: We really appreciate these constructive comments and agree that our initial draft was difficult to grasp. Unfortunately, in our work many already established acronyms come together - however, across the draft we removed the term "dICK" and exchanged it by the equally functional term "double-domain toxin" and therefore reduced the number of acronyms for clarity. We further improved the introduction in its second and fifth paragraph (Line 61 – 65 and 103 - 109 respectively),

where the relationships between the acronyms are introduced. It should be now clear to the reader that short cysteine-rich spider neurotoxins are ICK peptides, and that CSTX are a family within the spider ICK peptides. Within the CSTX, CPTX are a subgroup and they usually come as double-domain toxins (at least as per the literature opinion prior to our study). We do not (and never did) refer to any acronyms interchangeably and each always had its very distinct meaning. Of course, we are open to further clarifications if our modified version is still not reaching the desired clarity.

3) There is a full section describing CPTXs and PLA2s but not the other major venom components. Can the authors add an additional section describing these venom components and their function? This would add additional support for the function of the venom and provide necessary information to understand the selective pressures placed on the venom as a whole and not just the CPTXs.

Reply: We have expanded our discussion on the major venom components that important for the supposed defensive function and, most importantly, added more detail as to why they are very likely to act in this context (Line 167 - 195). We have also added a description of the activity spectrum of serine proteases as major putatively defensive element (Line 187 and following). During our modifications, we also realized that the membrane activity proposed for venom protein 11, which led us to hypothesize a potential defensive role (see main text for defensive criteria) is solely based on in silico inference and lacks experimental support. Hence, we reconsidered its function and removed it from the candidates considered important for defense in order to focus on components with sufficient functional evidence. This has been modified across the draft.

4) The last paragraph of the conclusion states “the most striking revelation was the difference between the targeting cell membranes to trigger rather than other spiders which target ion channels.” This was not described in this paper and the data generated in this paper do not support that claim. The “mode of action” and function of this venom was not studied and was not discussed in any detail in this manuscript. Can the authors remove these claims from the conclusion or heavily modify them and provide more context for these claims? The last sentence of this paragraph focuses more on the structure which was described in this manuscript. The authors should shift the focus on this paragraph to their own results.

Reply: Modified as suggested (Line 436 - 439).

Minor Points:

- Fourth Paragraph of the intro “The functionally most derived spider..” Needs a citation to back up claim. Could the authors also expand on how this venom is the “most derived”?

Reply: We have toned down the language here (Line 81 and thereafter) and rephrased large parts of the discussion to elaborate more on this matter (Line 167 – 195, line 336 – 340, line 365 - 368).

- “The two CSTX domains of CPTXs may be closely related, but further investigation is need to achieve a detailed understanding of their ecological role” I am unsure how looking at the relationship of these two domains can help understand their ecological role? Can the authors please rephrase this statement or expand on it?

Reply: This has been a mistake on our part, of course the suggested does not allow conclusions on ecological role or function. We intended to refer to “evolutionary relationships” and rephrased accordingly (Line 112).

- Last paragraph of the introduction: The authors talk about the structure and composition of *C. punctatorum* venom in relation to hymenopterans without a citation. This information needs to be supported by citations or moved into the discussion where the authors expand further.

Reply: We have removed the comment on the relation to hymenoptera venoms as we feel this is sufficiently discussed in our discussion and herein only refer to the broad statement of *C. punctatorum* toxins resembling known defensive toxins (Line 115 -122).

- Some of the percentages in the second paragraph of 2.1 do not align with the pie graph. For example, 40% of all venom transcripts are double domain CPTXs. However, the pie graph shows 58.1% were CPTXs with 8.4% of those are supposed to be single domain CPTXs. Another place is 12.7% are putative enzyme inhibitors, yet the graphs shows 12.6%. Can the authors verify the values in the graph and in the text?

Reply: The values have been double-checked and corrected if needed. We also chose some other toxins for the low amount components to discuss as their summing-up is more representative for the entire venom profile (the previously named toxins where all members of non-CSTX ICKs which amount to 0.3% of the venom but the text referred to their combined percentage with neuropeptide-like components and others resulting in a cumulative 4.9%).

- Figure 1: Can you split the insert in the middle to show single and double domain CPTXs? This might help to clarify some of the percentages shown above.

Reply: Modified as suggested.

- The abbreviation uCNTX is used without being described in full.

Reply: Section has been rephrased and this toxin family is no longer mentioned in the text (Line 165 – 197).

- Last paragraph of section 2.1: The authors state that the venom profile of *C. punctori* has predicted activities that are congruent with the painful symptoms. There are no citations to back this claim or the claim in the last sentence of section 2.1. Can the authors add more detail about these predicted activities with appropriate citations?

Reply: We have largely expanded this section and have devoted a distinct paragraph of 2.1 outlining why each of the components could be involved in defensive context (Line 165 -197). As outlined above, we also re-considered our classification for venom protein 11, which is not longer considered a defensive component based on the lack of functional evidence.

- Second paragraph in section 2.2. Why only test your hypothesis using CPTX and PLA2s? Why not use serine proteases and Venom Protein 11 as well? Can the authors provide a statement for why they focus on these two groups and not the others even though they are more abundant than the PLA2?

Reply: See our reply to major point 4).

- Figure 5: Can the authors use different colors for the three proteins? The different shades of red are hard to see differences between and it would be helpful to see how they line up in the overlay with more distinct colors.

Reply: Modified as suggested.

Reviewer #2 (Remarks to the Author):

Lüddecke and colleagues present a comprehensive characterization of the venom composition of the yellow sac spider *Cheiracanthium punctori*. Notably, this is the first report of a relatively high abundance of phospholipase A2 (PLA2) in a spider venom. The authors also demonstrate that the species' double-domain neurotoxins (CSTXs or CpTx) originated from a gene fusion event between two distinct single-domain toxin precursors, offering novel insight into the evolutionary mechanisms underpinning venom diversification.

These findings reveal a case of molecular convergence in venom systems across distantly related venomous animals (e.g., spiders, snakes, bees), suggesting that similar ecological pressures can lead to the recruitment of analogous molecular components across taxa.

After careful evaluation, we believe the manuscript is of high quality and merits publication. However, a number of points require clarification and refinement. Our specific suggestions are outlined below:

Reply: We thank the reviewer for the critical but constructive evaluation and overall positive feedback. We believe that their comments and concerns increased the quality of our manuscript and we happily included all of them to the best of our abilities. Please find below a point-by-point reply to each of the issues raised by the reviewer.

Major Comments

Functional classification of spider venoms:

Personally, I am not fully convinced by the binary classification of spider venoms into primarily “predatory” or “defensive” types. At least based on the evidence provided in this manuscript, I find it difficult to fully understand or accept this distinction. The authors argue that most spiders produce predatory venom, in part based on the observation that spider venoms generally do not cause significant harm to predators. However, if *C. puncturium* venom is to be classified as primarily defensive, does this imply that its venom has lost or diminished its capacity to paralyze prey? This assumption is not directly supported. Therefore, I tend to favor an alternative interpretation—namely, that spider venom primarily evolved for predation, but in some cases (such as *C. puncturium*), certain components have been secondarily enhanced to serve a defensive function. Unless the authors can provide more compelling evidence to support a redefinition of venom functionality, I suggest that expressions such as “defensive venoms” and “defensive spider venom” be used more cautiously or softened throughout the manuscript.

Reply: We agree with the reviewer that probably both functions are at play to some extent, which we aimed to express in our initial introduction. However, we acknowledge that our previous draft lacked clarity and we therefore rephrased parts of the introduction to better introduce the topic with its details and nuances (Line 68 -80). We explicitly state that most spider venoms are trophic but some components acquired defense as secondary function and that *C. puncturium* is the first of the latter that now underwent a more in-depth analysis.

Convergent evolution and dICK toxins:

The manuscript emphasizes convergent evolution of spider venom with that of other taxa in the title, particularly in the context of defensive function. However, a portion of the manuscript is devoted to the discussion of dICK toxins, without clearly establishing their contribution to pain induction or identifying which venom components from other lineages they might be functionally analogous to. For readers lacking a strong background in toxinology, this may obscure the connection between dICK toxins and the proposed convergent evolution. If the link between dICK toxins and defensive convergence is limited, the title or main text could benefit from minor revision to improve clarity for a broader readership.

Reply: We agree with the reviewer and have rephrased the title accordingly. Our title suggested that the entire venom arsenal of *C. puncturium* is convergently evolved to cause nociception. However, the case of convergence is only really apparent for the recruitment of PLA2 (convergence to defensive snake, insect, and scorpion venom) whilst for dICK evolution the case is not as clear. In agreement with the reviewer, we have rephrased the title and de-emphasized the focus on convergence in the sections where it was deemed necessary.

PLA2 abundance as a “major component”:

The authors assert that *C. puncturium* venom is the only known spider venom where PLA2 constitutes a major component. However, given that PLA2 has been identified in other spider venoms, the manuscript should cite or list comparative data from those species to establish a clear quantitative benchmark. This would help define the threshold at which PLA2 can be considered a “major component” in a meaningful context.

Reply: The reviewer is right that such a comparison would help to strengthen the manuscript. Hence, we modified this paragraph and discuss the matter in more detail, explicitly naming the number and,

if known, quantity of PLA2 identified in other spiders so far (Line 349 -369). Unfortunately, many past works are not quantitative and only report components on diversity level. However, our work shows that on protein-diversity level, *C. puncturium* contains ca. 75 times the PLA2 diversity of what is reported from other spiders. On the quantitative level, although based on a lower sample size, it contains 125 times as much PLA2. This provides a strong case for an increased amount of PLA2 in this functionally apotypic venom.

Minor Comments

Genomic localization of toxins:

The genome of *C. puncturium* has been published (GCA_038373885.1), albeit in a draft state. It would greatly strengthen the manuscript if the authors could determine the genomic loci of the identified toxin genes. This would add valuable genomic context to their evolutionary interpretations.

Reply: We agree with the reviewer here that such an analysis would help strengthen the findings. However, the currently available genome is only on scaffold level and does not allow a precise analysis for genomic localization. Furthermore, such an analysis would require the analysis of multiple taxonomically close and more distant groups to fully grasp the evolutionary processes involved. However, due to the quite limited selection of publicly available genomes (with useful annotation) at the needed quality, we believe that any such analysis in the current state would be highly biased and probably would lead to false conclusions. With that, we refrained from carrying out this analysis because we prefer to focus on the data available with the quality needed to answer a question. That said, we believe that future studies should investigate the genomic processes involved in spider toxin evolution – as such information is currently still lacking.

Venom gland depletion timing:

The methods state that venom glands were emptied at 144 h and 72 h before sampling. What was the rationale behind selecting these specific time intervals? Clarification of the biological or experimental considerations would be helpful.

Reply: Has been added to the section as suggested by the reviewer (Line 457 -463).

Duplicated CPTX2a label in Figure 3A:

There appear to be two entries labeled CPTX2a in Figure 3A. Please verify whether this is a labeling error.

Reply: The duplicated entry has been removed from the alignment.

CPTX14a–c categorization and Figure 1B consistency:

CPTX14a–c are classified as PLA2s in Figure 5A, but in Figure 1B, CPTX entries are listed separately and in parallel with PLA2s. Could this be a labeling inconsistency or error in nomenclature? Please double-check.

Reply: Based on the reviewer feedback, we discovered that our nomenclature for venom components from *C. puncturium* was too close to the nomenclature of CPTX-type CSTX toxins we discuss in our paper. Therefore, we renamed all non-CPTX components and modified accordingly. The new version of figure 5 should be much clearer now.

Color distinction in Figure 5B:

The color scheme used in Figure 5B results in models that are visually indistinguishable in the overlay image. Please consider using a more contrasting color palette to enhance clarity.

Reply: Modified as suggested.

Clarification on inference of independent evolution in Section 2.3:

In Section 2.3, the authors state: “The absence of a lone second domain in *C. puncturium* supports the independent evolution of the double domain within the family Cheiracanthiidae,...” Could the

authors clarify how the absence of a standalone second domain supports the hypothesis of independent evolution?

Reply: The mechanistic explanation to this statement follows in the next paragraph of that section but based on the reviewers comment, we feel as if our current formatting made it difficult to follow (i.e. the paragraphing was quite complicated and the textblocks were divided by a figure) We have restructured this paragraph (Line 258 -289) now and the explanation follows directly to the above statement. We feel as if the details and nuances of it are now much clearer but are happy to further elaborate if the reviewer and editor feel this is needed.

On the interpretation of purifying selection and venom system loss (Section 2.3):

In the third paragraph of section 2.3, the authors state:

“Spider venoms in general and CSTXs in particular are known to evolve under strong purifying selection, which regularly leads to the loss of complexity and even entire venom systems.”

I find this statement somewhat confusing. It is unclear to me how purifying selection—typically associated with the conservation of functional elements—could lead to the complete loss of an entire venom system. Perhaps I have misunderstood the intended meaning, but this point could benefit from further clarification in the manuscript.

Reply: This topic has been previously discussed intensively within the (spider) venom community. Yes, purifying selection tends to cause conservation of functional elements. However, venoms are metabolically expensive traits and tend to get lost easily once their function does not provide an economic advantage for the producing organism thanks to this strong influence of purifying selection. There are plenty examples for this from across the animal kingdom, for instance sea snakes that have lost their entire venom arsenal following a dietary transition from fish to fish-eggs where venom does not support its primary trophic function anymore (10.1007/s00239-004-0138-0). Among spiders, the primary examples are the reduced venoms of orb-weavers (Araneidae) that switched to a primarily web-based hunting and only secondary employ their venoms for prey capture. As a result, some araneid venoms became severely simplified and lost a large fraction of the unneeded venom toxins (see here for a discussion: <https://doi.org/10.3390/biom10070978>). Further, Uloboridae completely omitted their venom for hunting and evolved a very advanced web-based hunting strategy. In response, these spiders completely lost their venom system (<https://doi.org/10.1186/s12915-025-02248-1>). We have kept this section originally very short in order to keep the draft more concise. However, in response to this comment it became obvious to us that our previous phrasing was insufficient and we rephrased for clarity and further elaborated on this topic (Line 303 - 313).